# Virus-like Particles of Nodavirus Displaying the Receptor Binding Domain of SARS-CoV-2 Spike Protein: A Potential VLP-Based COVID-19 Vaccine

**DOI:** 10.3390/ijms24054398

**Published:** 2023-02-23

**Authors:** Kiven Kumar, Wen Siang Tan, Siti Suri Arshad, Kok Lian Ho

**Affiliations:** 1Department of Pathology, Faculty of Medicine and Health Sciences, Universiti Putra Malaysia, UPM, Serdang 43400, Selangor, Malaysia; 2Department of Microbiology, Faculty of Biotechnology and Biomolecular Sciences, Universiti Putra Malaysia, UPM, Serdang 43400, Selangor, Malaysia; 3Department of Veterinary Pathology and Microbiology, Faculty of Veterinary Medicine, Universiti Putra Malaysia, UPM, Serdang 43400, Selangor, Malaysia

**Keywords:** COVID-19 vaccine, receptor binding domain, virus-like particles, SARS-CoV-2, nodavirus, VLP-based vaccine

## Abstract

Since the outbreak of the coronavirus disease 2019 (COVID-19), various vaccines have been developed for emergency use. The efficacy of the initial vaccines based on the ancestral strain of severe acute respiratory syndrome coronavirus type 2 (SARS-CoV-2) has become a point of contention due to the emergence of new variants of concern (VOCs). Therefore, continuous innovation of new vaccines is required to target upcoming VOCs. The receptor binding domain (RBD) of the virus spike (S) glycoprotein has been extensively used in vaccine development due to its role in host cell attachment and penetration. In this study, the RBDs of the Beta (β) and Delta (δ) variants were fused to the truncated *Macrobrachium rosenbergii* nodavirus capsid protein without the protruding domain (CΔ116-*Mr*NV-CP). Immunization of BALB/c mice with the virus-like particles (VLPs) self-assembled from the recombinant CP showed that, with AddaVax as an adjuvant, a significantly high level of humoral response was elicited. Specifically, mice injected with equimolar of adjuvanted CΔ116-*Mr*NV-CP fused with the RBD of the β- and δ-variants increased T helper (Th) cell production with a CD8^+^/CD4^+^ ratio of 0.42. This formulation also induced proliferation of macrophages and lymphocytes. Overall, this study demonstrated that the nodavirus truncated CP fused with the SARS-CoV-2 RBD has potential to be developed as a VLP-based COVID-19 vaccine.

## 1. Introduction

The etiologic agent for coronavirus disease 2019 (COVID-19) is a novel coronavirus known as the severe acute respiratory syndrome coronavirus type 2 (SARS-CoV-2). Since the first COVID-19 cases reported in late 2019 in Wuhan City, China, the virus has spread rapidly to all parts of the world. To mitigate the virus transmission, scientists raced against time to develop COVID-19 vaccines using various platforms including mRNA, whole inactivated virus, and viral vectors, which were approved for emergency use by the World Health Organization (WHO) soon after declaration of COVID-19 as a pandemic. The spike (S) glycoprotein of SARS-CoV-2 of the original Wuhan strain was used to design the early vaccines. The efficacies of these vaccines were between 70 to 95% against SARS-CoV-2 infection [1,2]. However, continuous mutations which give rise to new variants have significantly affected the vaccine efficacies. The emergence of variants of concerns (VOCs), including Alpha (α), Beta (β), Gamma (γ), Delta (δ), and Omicron (ο), causes fresh waves of illnesses all over the world. For instance, the sera collected from BNT162b2 receivers [3,4,5,6,7,8,9] and mRNA-1273 recipients [9,10] showed significantly low levels of neutralizing antibodies against the β-variant, and the effectiveness of Novavax vaccine (NVX-CoV2372) and ChAdOx1, non-mRNA-based vaccines, was merely above 50% and 20%, respectively [11]. The δ-variant demonstrated a greater transmissibility, higher viral loads, and higher reinfection rates [12,13]. These variants have evaded the natural immunity of humans regardless of their vaccination status [14]. Collectively, the effectiveness of these vaccines is lower, as shown in the clinical trials of two adenovirus-based vaccines, a nanoparticle-based vaccine, and an inactivated protein-based vaccine against VOCs [15].

*Macrobrachium rosenbergii* nodavirus (*Mr*NV) is the pathogen that causes white-tail disease (WTD) in the giant freshwater prawn [16]. The viral genome consists of positive-sense single-stranded (+ssRNA) bipartite RNA molecules. The larger (3.1 kilobases) RNA molecule encodes the RNA-dependent RNA polymerase (RdRp) and the B2-like protein, while the smaller RNA molecule (1.2 kilobases) is translated to the viral capsid protein (CP) [17]. The full-length CP is divided into the N-terminal shell (S) domain consisting of 255 amino acid residues, while the smaller protruding (P) domain comprising of 116 amino acid residues towards the C-terminus of the protein [18]. *Mr*NV-CP can be produced in *Escherichia coli* or *Spodoptera frugiperda* (*Sf*9) insect cells, in which the recombinant protein self-assembles into virus-like particles (VLPs), morphologically resembling the native virus [17,19]. The VLPs were manipulated to display foreign epitopes including the hepatitis B virus (HBV) ‘a’ determinant [20], the influenza A virus (IAV) ectodomain of the matrix 2 (M2e) protein, and the Japanese encephalitis virus (JEV) envelope protein domain III [21], in which the epitopes were fused at the C-terminal region of the monomeric CP. These studies demonstrated the potential of *Mr*NV-CP to be employed as a nano-platform for VLP-based vaccine development. In addition, the mice immunized with the VLPs derived from *Mr*NV-CP harboring three copies of IAV M2e survived a lethal challenge with IAV subtypes H1N1 and H3N2 [22].

Structural analysis of the cryo-electron microscopy of *Mr*NV capsids revealed that the C-terminus of the full-length CP is located at the narrow linker between the protruding spikes formed by the P-domain and spherical shell formed by the S-domain [18,23]. The bulky blade-like spike of *Mr*NV-CP might prevent full display of foreign epitopes that are fused to the C-terminus of the CP. To maximize the exposure of the foreign epitope, the P-domain (116 residues) of the *Mr*NV-CP used in this study was genetically replaced with the RBDs derived from β- (CΔ116-*Mr*NV-CP^β-RBD^) and δ-SARS-CoV-2 (CΔ116-*Mr*NV-CP^δ-RBD^), after which the proteins were expressed in *E. coli*. The immunogenicity, immunophenotyping, and cytokine profiles of CΔ116-*Mr*NV-CP^β-RBD^ and CΔ116-*Mr*NV-CP^δ-RBD^ were studied in BALB/c mice. The results revealed that the BALB/c mice immunized with a mixture of VLPs derived from CΔ116-*Mr*NV-CP^β-RBD^ and CΔ116-*Mr*NV-CP^δ-RBD^, as well as in the presence of the AddaVax adjuvant, mounted significant levels of innate and humoral immune responses. These findings demonstrated that the chimeric VLPs derived from the mixture of CΔ116-*Mr*NV-CP^β-RBD^ and CΔ116-*Mr*NV-CP^δ-RBD^ have potential to be further developed into a VLP-based SARS-CoV-2 vaccine.

## 2. Results

### 2.1. Plasmid Construction, Protein Expression, and Purification

The coding sequence of the RBDs derived from β- or δ-SARS-CoV-2 (~600 bp), containing the EcoRI and HindIII cutting sites at the 5′ and 3′ ends, respectively, was synthesized. The restriction enzyme cutting sites were introduced to enable ligation of chemically synthesized RBDs to the coding sequence of the P-domain-truncated *Mr*NV-CP (CΔ116-*Mr*NV-CP) (Appendix A). The resulting recombinant plasmids, pCΔ116-*Mr*NV-CP^β-RBD^ and pCΔ116-*Mr*NV-CP^δ-RBD^, encode the fusion proteins containing 481 and 482 amino acid residues, respectively (Figure 1a). The recombinant plasmids were introduced into *E. coli* BL21 (DE3) competent cells, after which isopropyl β-D-1-thiogalactopyranoside (IPTG) was used to induce protein production. The CΔ116-*Mr*NV-CP, which served as a control, was purified as a ~33 kDa protein (Figure 1b, lane 1), whereas the CΔ116-*Mr*NV-CP^β-RBD^ and CΔ116-*Mr*NV-CP^δ-RBD^ were purified as recombinant proteins with a molecular mass (M_r_) of ~55 kDa (Figure 1b, lanes 2 and 3, respectively). This M_r_ corresponded well with the calculated M_r_ of CΔ116-*Mr*NV-CP^β-RBD^ and CΔ116-*Mr*NV-CP^δ-RBD^ of 54.49 kDa and 54.64 kDa, respectively. In Western blot analysis, the recombinant proteins CΔ116-*Mr*NV-CP^β-RBD^ and CΔ116-*Mr*NV-CP^δ-RBD^, probed by the anti-His monoclonal antibody, also gave rise to a single protein band of ~55 kDa (Figure 1c, lanes 2 and 5). Other recombinant proteins used in this study, CΔ116-*Mr*NV-CP (~33 kDa), β-RBD (~21 kDa), Wuhan-SARS-CoV-2-RBD (~21 kDa), and δ-RBD (~21 kDa), were purified using immobilized metal affinity chromatography (IMAC), and detected with the anti-His monoclonal antibody in the Western blotting (Figure 1c, lanes 1, 3, 4, and 6, respectively).

### 2.2. Scanning Transmission Electron Microscopy (STEM)

STEM analysis showed that CΔ116-*Mr*NV-CP, CΔ116-*Mr*NV-CP^β-RBD^, and CΔ116-*Mr*NV-CP^δ-RBD^ self-assembled into icosahedral VLPs with diameters ranging from ~18 to ~20 nm. The CΔ116-*Mr*NV-CP without the RBD was used as a control (Figure 2a). Both the VLPs of CΔ116-*Mr*NV-CP^β-RBD^ and CΔ116-*Mr*NV-CP^δ-RBD^ appeared similar in size and shape (Figure 2b,c).

### 2.3. Dynamic Light Scattering (DLS) Analysis

DLS analysis indicated that the average particulate sizes of the VLPs formed by CΔ116-*Mr*NV-CP (Figure 3a), CΔ116-*Mr*NV-CP^β-RBD^ (Figure 3b), and CΔ116-*Mr*NV-CP^δ-RBD^ (Figure 3c) were 16.99, 18.40, and 18.39 nm, respectively. The polydispersity indices of the VLPs formed by CΔ116-*Mr*NV-CP, CΔ116-*Mr*NV-CP^β-RBD^, and CΔ116-*Mr*NV-CP^δ-RBD^ were 0.59, 0.25, and 0.29%, respectively.

### 2.4. Immunogenicity of the Chimeric VLPs

After one week of acclimatization, the mice were immunized subcutaneously at weeks 2 (first injection), 5 (first booster), and 8 (second booster) with adjuvanted or unadjuvanted (i) CΔ116-*Mr*NV-CP, (ii) CΔ116-*Mr*NV-CP^β-RBD^, (iii) CΔ116-*Mr*NV-CP^δ-RBD^, (iv) a mixture of VLPs (CΔ116-*Mr*NV-CP^β-RBD^ and CΔ116-*Mr*NV-CP^δ-RBD^ in equal concentrations; Mix-VLPs), and (v) HEPES buffer (negative control). Sera were collected from the mice prior to each injection, and ELISA was performed to determine the antibodies against the RBDs. Generally, anti-RBD antibodies were not detected in the sera collected before immunization with the chimeric VLPs (Figure 4). Following the first injection with either the chimeric VLPs of CΔ116-*Mr*NV-CP^β-RBD^ or CΔ116-*Mr*NV-CP^δ-RBD^ or the Mix-VLPs, regardless of the presence or absence of the AddaVax adjuvant, antibodies against the β- (Figure 4b) and the δ-SARS-CoV-2 RBD (Figure 4c) were detected. The anti-β- and anti-δ-SARS-CoV-2 RBD antibodies increased by about two folds after the first booster. Subsequently, the second booster further increased the antibody levels.

Interestingly, these antibodies also interacted with the RBD derived from the ancestral Wuhan strain (Figure 4a), although the level of antibodies was only one third of the total antibody captured by the RBDs derived from β- and δ-SARS-CoV-2. With respect to the negative controls, anti-RBD antibodies were not detected in the sera collected from the mice injected with HEPES or VLPs of CΔ116-*Mr*NV-CP with or without the adjuvant. The test group inoculated with the Mix-VLPs produced higher levels of anti-β- and anti-δ-SARS-CoV-2 RBD antibodies when compared with the groups immunized separately with CΔ116-*Mr*NV-CP^β-RBD^ and CΔ116-*Mr*NV-CP^δ-RBD^ (Figure 4b,c).

### 2.5. Determination of Mouse Splenocytes by Immunophenotyping

Mouse spleens collected one week after the second booster (week 9) were used to analyze the presence of cytotoxic T lymphocyte (CTL; CD3^+^ and CD8^+^) and T helper (Th; CD3^+^ and CD4^+^) cells. The highest percentage of CD3^+^CD4^+^ cell population was observed in the splenocytes isolated from the mice administered with the Mix-VLPs regardless of the presence of adjuvant, followed by those from the test groups inoculated with the VLPs of CΔ116-*Mr*NV-CP^β-RBD^ and CΔ116-*Mr*NV-CP^δ-RBD^ with or without the adjuvant (Table 1). The negative control groups inoculated with CΔ116-*Mr*NV-CP, regardless of the presence of adjuvant, also triggered proliferation of significantly high levels of CD3^+^CD4^+^ and CD3^+^CD8^+^ compared to the buffer-inoculated mice. Figure 5a shows that the CD8^+^/CD4^+^ T-cell ratio was significantly altered in the test groups immunized with the VLPs of CΔ116-*Mr*NV-CP^β-RBD^ or CΔ116-*Mr*NV-CP^δ-RBD^ or the Mix-VLPs, with or without adjuvant, compared to the negative control groups. Additionally, high levels of macrophage population were also detected in animal groups immunized with the VLPs of CΔ116-*Mr*NV-CP^β-RBD^ or CΔ116-*Mr*NV-CP^δ-RBD^ or the Mix-VLPs, with or without the adjuvant, whereas such macrophage elevation was not observed in the negative control groups (Figure 5b).

### 2.6. Quantification of Cytokines

Cytokine quantification with multiplex ELISA showed that the mice immunized with the chimeric proteins exhibited significant elevations in all the serum cytokines tested, except IL-6 (Figure 6). The increments were more obvious in the animal groups immunized with the VLPs of CΔ116-*Mr*NV-CP^β-RBD^ or CΔ116-*Mr*NV-CP^δ-RBD^ or Mix-VLPs, with or without the adjuvant, than in the negative control groups inoculated with HEPES buffer or the VLPs of CΔ116-*Mr*NV-CP, with or without the adjuvant. The result indicated that the presence of SARS-CoV-2 RBD on the surface of nodavirus VLPs, regardless of the variants, could elicit the production of IL-5, IL-12p70, IFN-γ, and TNFα. Figure 6 shows the cytokine profile of dendritic cell (DC), CTL, and macrophage activities. Mice immunized with the Mix-VLPs, with or without adjuvant, induced the production of Th1 cytokines [IFN-γ (Figure 6a), IL-12p70 (Figure 6b) and TNF-α (Figure 6c)], Th2 eosinophil cytokine [IL-5 (Figure 6d)] and macrophage and DC-secreted cytokines [IL-6 (Figure 6e)]. Likewise, mice that received the VLPs of CΔ116-*Mr*NV-CP^β-RBD^ or CΔ116-*Mr*NV-CP^δ-RBD^, with or without adjuvant, also effectively induced the cytokines effectively compared to mice injected with the buffer and the VLPs formed by CΔ116-*Mr*NV-CP, with or without adjuvant. The result also revealed that both pro-inflammatory Th1 (IFN-γ, IL-12p70, and TNF-α) and anti-inflammatory Th2 (IL-5) cytokines were elevated following administration of VLPs displaying the RBDs of the β- or δ-variants.

## 3. Discussion

The COVID-19 pandemic has resulted in substantial morbidity and death across the globe. Multiple variants have emerged due to the high mutability and transmissibility of the virus. The original wild-type strain isolated in Wuhan, China, and its subsequent variants with higher transmissible rates have heightened the need for global vaccinations. Various types of vaccines have been speedily developed to prevent further spread of the virus and reduce the severity of the disease. Current SARS-CoV-2 vaccines available in the market include the mRNA-based vaccines (Comirnaty and Spikevax), viral-vector-based vaccines (COVISHIELD and Ad26.COV2.S), and inactivated virus vaccines (CoronaVac and Covaxin). The effectiveness of these vaccines was reported to be up to 95% [1,24,25]. However, there have been concerns around the world about the lower effectiveness against new variants, particularly VOCs [26]. The first VOC, namely the Alpha variant (α-SARS-CoV-2, known as B.1.1.7 variant), was identified in the UK. Alpha-SARS-CoV-2 contains an amino acid substitution of D614G, a point mutation on the viral spike glycoprotein [27]. This variant is more transmissible than the wild-type Wuhan strain, and it has caused a greater rate of mortality in the UK [27]. Besides the α-variant, the vaccine effectiveness of the ChAdOx1 nCoV-19 vaccine against the β-variant (lineage B.1.351 isolated in South Africa) dropped from 89.3% to 21.9%, and, in separate studies, collectively, the vaccine effectiveness of it dropped to 62% [11,28]. Another study showed that the vaccine effectiveness of Novavax vaccine (NVX-CoV2373) against the B.1.1.7 and B.1.351 variations dropped to 85.6% and 60%, respectively, in a phase III trial [29]. These data attest the decreasing efficacy of these licensed COVID-19 vaccines against the new variants.

In this study, the P-domain of the full-length of *Mr*NV-CP was removed by deleting 116 amino acid residues from its C-terminus. Removal of the P-domain allows the fusion peptide to be efficiently displayed on the surface of the VLPs formed by the truncated capsid protein, CΔ116-*Mr*NV-CP. The RBDs of β- and δ-VOCs were separately displayed on the outer surface of the chimeric VLPs assembled from CΔ116-*Mr*NV-CP^β-RBD^ and CΔ116-*Mr*NV-CP^δ-RBD^, respectively. The results demonstrated that these chimeric proteins, with M_r_ of ~52 kDa, were successfully expressed and purified, as analyzed with SDS-PAGE. STEM analysis revealed that the diameters of the VLPs formed by these chimeric proteins were similar to those of CΔ116-*Mr*NV-CP. Several studies showed that short peptides, such as the IAV M2e, HBV ‘a’ determinant, and GE11 peptide that fused to the full-length *Mr*NV-CP, had no effect on the diameter of the VLPs formed [20,30,31]. Our previous study showed that the fusion of the domain III of the Japanese encephalitis virus (JEV) envelope protein, consisting of 133 amino acid residues at the C-terminus of the full-length *Mr*NV-CP, drastically reduced the diameter of the VLPs from 30 to 18 nm [21]. In the present study, the VLPs formed by CΔ116-*Mr*NV-CP^β/δ-RBD^ also demonstrated a similar reduction in diameter to ~18 nm. Nevertheless, it is still unsure how the foreign epitopes are displayed on the CΔ116-*Mr*NV-CP. A high-resolution three-dimensional structure of these chimeric VLPs would provide insights into this enigma.

In the immunogenicity assay, ELISA was used to determine the level of anti-RBD antibodies induced by the VLPs of CΔ116-*Mr*NV-CP^β-RBD^ and CΔ116-*Mr*NV-CP^δ-RBD^ and the Mix-VLPs in the presence or absence of the AddaVax adjuvant. The antisera of mice immunized with the chimeric VLPs were captured by the RBDs of the Wu-, β-, and δ-strain coated on the microtiter plate wells. The result showed that the antibody specific against the β-SARS-CoV-2 RBD was elicited in mice either immunized with the VLPs of CΔ116-*Mr*NV-CP^β-RBD^ or the Mix-VLPs with or without the AddaVax adjuvant. Similarly, the antibody against the δ-SARS-CoV-2 RBD was also produced in mice inoculated with the VLPs of CΔ116-*Mr*NV-CP^δ-RBD^ or the Mix-VLPs, regardless of the presence or absence of the AddaVax adjuvant. Interestingly, the Wu-RBD coated on the plate showed a lower affinity to antisera elicited by the VLPs displaying the β-RBD or δ-RBD and the Mix-VLPs with or without the adjuvant. This result indicated that the anti-RBD antibodies produced in mice immunized with the VLPs of CΔ116-*Mr*NV-CP^β-RBD^ and CΔ116-*Mr*NV-CP^δ-RBD^ demonstrated a lower affinity towards the Wu-SARS-CoV-2 RBD.

Numerous studies have proved that SARS-CoV-2 RBDs can evade the hosts’ immune systems and enhance antibody neutralization resistance [32,33,34,35]. VOCs such as β- and δ-variants, derived from their ancestral Wuhan strain (Hu-1), possess greater transmissibility, and enhanced disease severity [36]. Specifically, reduced neutralization was reported from individuals who had been fully vaccinated or recovered from the Wu-strain [37,38,39]. Other studies showed that the neutralizing antibodies against the β- and δ-variants reduced by 2.7 and 1.4 times, respectively [4,5,40,41]. In addition, the neutralizing antibody titers reduced by 10.4, 2.3, and 2.1 folds against the β-, γ-, and δ-variants, respectively, when compared with that of the ancestral virus in fully vaccinated healthcare workers [5,10]. As reported by Cherian et al. [42], the β- and δ-strains contain three (K417N, E484K, and N501Y) and two (T478K and L452R) point mutations within their RBD regions, respectively, when compared to the Wu-strain. These mutations may result in partial circumvention of the neutralizing humoral immunity established by spontaneous infection or vaccination, resulting in reinfections or infections [43,44].

A good understanding of the T cell responses towards the chimeric VLPs is important to develop effective COVID-19 vaccines. In this study, the fusion of RBDs at the C-terminal end of CΔ116-*Mr*NV-CP enhanced the Th cell population. The Mix-VLPs in the presence of the AddaVax adjuvant showed the highest level of CD4^+^ Th cells. At the same time, the level of CTLs showed no significant difference compared with that of other study groups except for the negative control groups inoculated with the HEPES buffer or the VLPs of CΔ116-*Mr*NV-CP with or without the adjuvant. Most of the immunized groups showed activation of CD4^+^ Th activity. Previous studies using the VLPs formed by the full-length *Mr*NV-CP as nano-carriers for various epitopes showed that the activation of CD4^+^ cells was common compared to CD8^+^ cells [20,30,45]. Similar observations were also reported in SARS-CoV-2 infection, which corresponded well with the ability of CD4^+^ T cell immune responses in suppressing initial SARS-CoV-2 infections [46,47,48]. In managing SARS-CoV infection in animal models, CD4^+^ T cells were noted to be more predominant than CD8^+^ T cells [49,50]. Compared to CD4^+^ and CD8^+^ T cells, SARS-CoV-2-specific CD4^+^ T cells showed a strong correlation with reduced COVID-19 severity [47]. In acute COVID-19 cases, rapid activation of SARS-CoV-2-specific CD4^+^ T cells was shown to cause milder symptoms with faster viral clearance [51]. On the other hand, the lack of CD4^+^ T cells specific to SARS-CoV-2 has been linked with more severe cases (>day-22 post-symptom onset in some patients) [47,51,52]. Due to their capacity to destroy virus-infected cells, CD8^+^ T cells are essential for viral clearance.

Macrophages are among the first immune cells to interact with viruses that invade the human body. In the SARS-CoV-2-induced inflammatory response, monocytes and macrophages play a critical role in clearing the virus by the innate immune response [53]. The present study showed that macrophage activation occurred in all the immunized mouse groups except the negative control group inoculated with the HEPES buffer. Theobald et al. [54] reported that the SARS-CoV-2 S protein triggers macrophage activation. Interestingly, the activation of macrophages depends on the types of VLPs. Han et al. [55] reported that norovirus VLPs could not efficiently increase dendritic cell (DC) maturation and antigen presentation, but they are capable of activating macrophages. Another study by Lenz et al. [56] demonstrated that the papillomavirus VLPs could induce activation of DCs. Overall, activation of macrophages or DCs is antigen-receptor-dependent, which facilitates the binding and uptake of VLPs [57].

Blanco-Melo et al. [58] demonstrated a different inflammatory response associated with SARS-CoV-2 infection in COVID-19 patients. Individuals with comorbidities are more likely to have an “inappropriate and inadequate immune response”. This might encourage viral replication and exacerbate symptoms associated with illness severity [58]. IFN-γ is a type II interferon generated by lymphocytes such as CD4^+^ and CD8^+^ T cells, Treg cells, CD8 T cells, FoxP3^+^, NK cells, and B cells [59]. Thus, IFN-γ is a central antiviral immune mediator. A study indicated that the blood IFN-γ level was greater in COVID-19 patients than that in healthy individuals. The increased levels of this and other cytokines could likely be due to the activation of Th1 and Th2 cells [60]. In this study, a high titer of IFN-γ was detected in the mice inoculated with the Mix-VLPs with AddaVax adjuvant, followed by those immunized with this formulation without the adjuvant. This is likely due to the presence of the two RBDs from β- and δ-SARS-CoV-2. Sun et al. [61] showed that the IFN-armed RBD dimer enhanced the production of IFN-γ by the CD4^+^ T cells. In the same study, the mice immunized with monomeric RBD gave rise to lower IFN-γ compared to the dimeric RBD.

DCs and macrophages release IL-12p70 in response to microbial stimuli such as viral infections. In combination with IL-15, IL-18, and type I IFN, the cytokine increases NK cells’ cytotoxicity and causes IFN secretion [62]. In this study, all the adjuvanted study groups except CΔ116-*Mr*NV-CP with adjuvant showed high levels of IL-12p70. Several studies showed that the adjuvanted vaccines enhanced the stimulation of IL-12p70 [7,63]. Moreover, IFN-γ, IL-12p70, and TNF-α were stimulated by most of the COVID-19 vaccines, indicating the activation of these cytokines by the viral proteins, particularly the S protein [64,65,66].

Inflammation is the initial innate immune response triggered by pro-inflammatory cytokines or chemokines. Therefore, excessive production of these pro-inflammatory molecules during an infection could lead to an overwhelmed immune response known as the cytokine storm, which may lead to acute respiratory distress syndrome (ARDS) or multiple organ failure. TNF-α and IL-6 are commonly involved in inflammation during SARS-CoV-2 infection [67]. The frequency of specific TNF-α induced by the Mix-VLPs with the adjuvant in the current study is similar to that reported in clinical trials of COVID-19 vaccines such as SARS-CoV-2 FINLAY-FR-1A dimeric-RBD recombinant vaccine and recombinant spike protein nanoparticle vaccine [68,69]. A mixture of VLPs with the adjuvant enhanced the population of TNF-α-producing effector T cells, indicating induction of cellular immunity. COVID-19 mitigation relies heavily on T cell responses. CD4^+^ T cells are effector cells that produce IFN-γ, TNF-α, and other cytokines, besides collaborating with B cells [70,71]. TNF-α released by effector T cells and innate immunity cells can destroy virus-infected cells. Multiplication of T cells, cytokine generation, and host survival are all aided by the cellular responses [68]. 

After immunization with either the VLPs of CΔ116-*Mr*NV-CP^β-RBD^ or CΔ116-*Mr*NV-CP^δ-RBD^ or the Mix-VLPs, the mice secreted more IFN-γ, IL-12p70, and TNF-α than IL-5 or IL-6, suggesting a Th1-dominant response. Although IL-5 and IL-6 were elicited, their titers were lower than the Th1 cytokines. Overall, immunization with the Mix-VLPs in the presence of the AddaVax adjuvant elicited a higher Th1 response than Th2 response in BALB/c mice.

## 4. Materials and Methods

### 4.1. Construction of Truncated MrNV-CP without the Protruding Domain (CΔ116-MrNV-CP)

The coding fragment of *Mr*NV-CP without the P-domain (CΔ116-*Mr*NV-CP) was amplified from the plasmid pTrcHis-TARNA2, containing the coding sequence of full-length *Mr*NV-CP [17]. The primers used in PCR are as follows: Forward primer: 5′-GGGTAAACCATGGCCCTTAACATCAAGATGGCTAGAGGTAAA-3′ (underlined sequence indicates NcoI restriction site), and reverse primer: 5′-TTTTTGAATTCGCCCTTCCCTAACTGTGAAATTTCCACTGGTGT-3′ (underlined sequence indicates EcoRI restriction site). The PCR reaction was carried out with the Velocity DNA polymerase (Bioline, London, UK). Initially, the DNA was denatured at 98 °C for 5 min, followed by DNA amplification with 35 cycles of denaturation, annealing, and extension at 98 °C for 30 s, 61 °C for 1 min, and 72 °C for 1 min, respectively, and completed with a final extension at 72 °C for 10 min. Both the purified PCR product and plasmid pTrcHis2 TOPO were digested with NcoI and EcoRI restriction enzymes and purified using the Gel Cleanup Kit (Qiagen, Hilden, Germany) based on the recommended protocol. Ligation of the digested pTrcHis2 TOPO plasmid and purified PCR product was performed with T4 DNA ligase (Promega, Madison, WI, USA) overnight at 4 °C. The ligated plasmid was introduced into *E. coli* TOP10 competent cells (Thermo Fisher Scientific, Waltham, MA, USA) using the heat-shock method.

### 4.2. Expression and Purification of Chimeric VLPs

The expression of the CΔ116-*Mr*NV-CP was adapted from a previous study [17]. *E. coli* containing the recombinant plasmid was grown in Luria Bertani (LB) broth containing ampicillin (100 µg/mL) overnight at 37 °C. The overnight culture (20 mL) was transferred into LB broth (1 L), and cultured at 37 °C at 200 rpm until the OD_600_ of the culture reached ~0.6. The culture’s temperature was lowered to 25 °C before adding 1 mM IPTG, and the culture was incubated for 5 h. The bacterial cells were recovered by centrifugation at 8000× *g* for 10 min, and the protein was purified as described by Goh et al. [17].

### 4.3. Synthesis of the Coding Sequences of the Wuhan, Beta, and Delta Variants of the SARS-CoV-2 RBD

The DNA fragments containing the SARS-CoV-2 RBD coding sequences of the ancestral Wuhan strain and β- and δ-VOCs were synthesized. The RBD coding sequence of the Wuhan strain was obtained from 2019-nCoV WHU01 (GenBank accession no: MN988668.1), whereas the coding sequences of the β- and δ-variants were obtained from hCoV-19/Malaysia/IMR_WC75452/2021 (GISAID accession no: 1263540) and hCoV-19/Malaysia/IMR_035575/2021 (GISAID accession no: 2931924), respectively. To enable ligation of the synthesized Wuhan and β-variant RBD coding sequences to the plasmid containing the coding sequence of CΔ116-*Mr*NV-CP, EcoRI and HindIII restriction sites were included at the 5′- and 3′-ends, respectively, whereas the coding fragment of the δ-variant RBD was designed to contain HindIII and the SnaBI recognition sequence at the 5′- and 3′-ends, respectively. The synthesized DNA fragments were ligated into the pUCIDT vector and stored in TE buffer at a final concentration of 4 µg/µL. The plasmid was introduced into *E. coli* BL21 (DE3) (Thermo Fisher Scientific, Waltham, MA, USA) competent cells.

### 4.4. Ligation of the SARS-CoV-2 RBD Coding Fragments to the pTrcHis2-TOPO Vector

The coding fragments synthesized as mentioned above were ligated into the pTrcHis2 TOPO TA expression vector. Firstly, the coding fragments were amplified by PCR using the proofreading polymerase; Velocity DNA Polymerase (Meridian Bioline, London, UK). The PCR primers used to amplify these coding fragments are listed in Table 2. The PCR reactions were performed as described in Section 4.1 except the annealing temperature was changed to 59 °C. The PCR products were then purified with the QIAquick PCR Purification Kit (Qiagen, Hilden, Germany), ligated as described above before being introduced into *E. coli* BL21 (DE3) (Thermo Fisher Scientific, Waltham, MA, USA) competent cells for protein expression.

### 4.5. Expression and Purification of Beta, Delta, and Wuhan-SARS-CoV-2 RBDs

The positive clones harboring the plasmids encoding the β-, δ-, and Wu-SARS-CoV-2 RBDs were cultured in 50 mL of LB broth supplemented with ampicillin (100 µg/mL) overnight at 37 °C. An aliquot of the overnight culture (10 mL) was transferred to 1 L LB broth containing ampicillin (100 µg/mL) in a 2 L conical flask. The culture was incubated at 37 °C at 200 rpm until the OD_600_ of the culture reached ~0.6. Protein expression was induced by adding IPTG (1 mM) for 5 h at 37 °C. Purification of β-, δ-, and Wu-SARS-CoV-2 RBDs was performed as described by Kumar et al. [21].

### 4.6. Construction of Plasmids Encoding the Beta Variant of SARS-CoV-2 RBD

The positive transformants containing the plasmids encoding the β-SARS-CoV-2 RBD were cultured in LB broth supplemented with ampicillin (100 µg/mL) at 37 °C for 18 h. The plasmids were extracted using the QIAprep Spin Miniprep Kit (Qiagen, Hilden, Germany) according to the standard protocol. Plasmids encoding the CΔ116-*Mr*NV-CP and the DNA fragment of β-SARS-CoV-2 RBD were linearized with EcoRI and HindIII, purified, and ligated with T4 DNA ligase. The ligated plasmid was transformed into *E. coli* BL21 (DE3) (Thermo Fisher Scientific, Waltham, MA, USA) competent cells via the heat-shock transformation method.

### 4.7. Construction of Plasmid Encoding the Delta Variant of SARS-CoV-2 RBD

The positive transformants carrying the plasmid encoding the δ-SARS-CoV-2 RBD were grown in LB broth supplemented with ampicillin (100 µg/mL), and incubated at 37 °C for 18 h. The recombinant plasmid was isolated with the QIAprep Spin Miniprep Kit (Qiagen, Hilden, Germany) according to the standard protocol. The coding region of the δ-SARS-CoV-2-RBD-containing restriction enzyme recognition sites was amplified using the forward primer (5’-AGGGCGAATTCGATGGTGGCGGAAATATTACAAACT-3’; underlined sequence indicates EcoRI restriction site), and the reverse primer (5′TACGTAAGCTTCAACAGTTGCTGGTGCATGTAGAAGTTCAAAA-3’; underlined sequence indicates HindIII restriction site). The PCR reaction contained Velocity DNA polymerase (0.5 U, 0.5 µL), dNTP mix (0.2 mM, 0.25 µL), 5× Hi-Fi reaction buffer (5 µL), forward and reverse primers (10 μM, 0.5 µL), and nuclease-free water was added to a final volume of 25 µL. PCR reactions were performed as described in Section 4.1 except that the annealing temperature was changed to 60 °C. The PCR product and plasmid pTrcHis2 TOPO encoding CΔ116-*Mr*NV-CP were digested with EcoRI and HindIII restriction enzymes, ligated, and introduced into *E. coli* BL21 (DE3) competent cells with the heat-shock transformation method.

### 4.8. Expression and Purification of Truncated MrNV-CP Fused with Beta or Delta SARS-CoV-2 RBDs

*Escherichia coli* cells carrying the recombinant plasmids encoding for CΔ116-*Mr*NV-CP^β-RBD^ or CΔ116-*Mr*NV-CP^δ-RBD^ were inoculated into 50 mL LB broth containing ampicillin (100 μg/mL) and cultured overnight at 37 °C. An aliquot (10 mL) of overnight culture was transferred to 1 L LB broth and incubated at 37 °C until the culture reached an OD_600_ of ~0.6. Protein expressions were induced by adding 1 mM IPTG into the culture, and it was incubated at 25 °C for 5 h. The cell pellets were harvested by centrifugation at 8000× *g* for 8 min. Purification of recombinant proteins was modified from a previous study by Kumar et al. [21], using a fast-protein liquid chromatography (FPLC) system (Äkta Purifier; GE Healthcare, Uppsala, Sweden). The HiTap^TM^ SP HP 1 mL column (GE Healthcare, Buckinghamshire, UK) was washed with 10 column volume (CV) of binding buffer (50 mM HEPES, 100 mM NaCl, pH 7.4). The lysates were loaded onto the column, and bound proteins were eluted via a gradient of NaCl concentration (50 mM HEPES, 1 M NaCl, pH 7.4) at a flow rate of 1 mL/min and fractionated using a Frac-950 collector. The eluted proteins were then analyzed with SDS-PAGE and Western blotting.

### 4.9. SDS-Polyacrylamide Gel Electrophoresis and Western Blotting

Purified proteins were analyzed with SDS-PAGE and Western blotting. The expression of CΔ116-*Mr*NV-CP, CΔ116-*Mr*NV-CP^β-RBD^, and CΔ116-*Mr*NV-CP^δ-RBD^ was confirmed with western blotting using the anti-His monoclonal antibody (1:5000 dilution in TBS; MERCK, Germany) as the primary antibody and the goat anti-mouse monoclonal antibody conjugated with alkaline phosphatase (1:5000 dilution in TBS; Santa Cruz Biotechnology, Dallas, TX, USA) as the secondary antibody.

### 4.10. Scanning Transmission Electron Microscopy (STEM)

Purified protein samples at an appropriate concentration (~0.4 mg/mL) were applied onto copper grids for 5 min before the grids were negatively stained with 2% (*w*/*v*) uranyl acetate for 6–8 min. The grids were air-dried completely before examining under a scanning transmission electron microscope (Cold FESEM, Regulus 8230, Hitachi Co., Tokyo, Japan).

### 4.11. Dynamic Light Scattering (DLS)

The hydrodynamic radius (R_h_) and homogeneity of the VLPs of CΔ116-*Mr*NV-CP, CΔ116-*Mr*NV-CP^β-RBD^, and CΔ116-*Mr*NV-CP^δ-RBD^ were measured with a Zetasizer Nano ZS (Malvern, Worcestershire, UK) using a 25-mW solid state laser at a wavelength of 780 nm at 24 °C.

### 4.12. Immunization of BALB/c Mice

Animal usage in this study was approved by the Institutional Animal Care and Use Committee (IACUC), Universiti Putra Malaysia (AUP No: R024/2021). A total of 72 female 5-week-old BALB/c mice were split into nine groups, consisting of 8 mice per group (n = 8). After one week of acclimatization, the mice were bled from the submandibular vascular bundles on either side of their cheeks with a 26G needle (BD Biosciences, Lakes, San Jose, CA, USA). Immunization was carried out by injecting 100 µL of 0.34 mg/mL of (i) CΔ116-*Mr*NV-CP (group 1), (ii) CΔ116-*Mr*NV-CP supplemented with AddaVax (InvivoGen, San Diego, CA, USA) with a 2:1 ratio (group 2), (iii) CΔ116-*Mr*NV-CP^β-RBD^ (group 3), (iv) CΔ116-*Mr*NV-CP^β-RBD^ supplemented with AddaVax (group 4), (v) CΔ116-*Mr*NV-CP^δ-RBD^ (group 5), (vi) CΔ116-*Mr*NV-CP^δ-RBD^ supplemented with AddaVax (group 6), (vii) the Mix-VLPs [a mixture of CΔ116-*Mr*NV-CP^β-RBD^ and CΔ116-*Mr*NV-CP^δ-RBD^ (ratio 1:1)] (group 7), (viii) the Mix-VLPs supplemented with AddaVax (group 8), (ix) and HEPES buffer, pH 7.4 (group 9). The first and second boosters were given on the fifth and eighth weeks, respectively. Blood samples (~100 µL) were withdrawn via submandibular bleeding from each mouse on the second, fifth, and eighth weeks before recombinant proteins were given. To obtain clear serum, the blood samples were stored at room temperature for 1 h before centrifugation at 3000× *g* for 15 min. The supernatant was transferred to a centrifuge tube and centrifuged again as above and stored at −80 °C.

### 4.13. Immunophenotyping of Mouse Splenocytes

Mouse spleens were harvested at week 9 and used for immunophenotyping. In brief, PBS was added to the spleens before the samples were meshed through a cell strainer (BD Biosciences, Franklin Lakes, San Jose, CA, USA). To obtain single-cell suspensions, the meshed samples were centrifuged at 300× *g* for 10 min before adding erythrocyte lysis (EL) buffer (155 mM NH_4_Cl, 10 mM KHCO_3_, 0.1 mM EDTA; pH 7.4) to resuspend the cell pellet for 10 min at 4 °C. The cells were then washed with PBS (10 mL) and resuspended in PBS supplemented with 1% (*w*/*v*) BSA. The supernatant was discarded, and EL buffer was added to the cell pellet until a whitish pellet was obtained. The pellet was then gently resuspended in ice-cold PBS-BSA (1 mL).

Using a hemocytometer, approximately 5 × 10^6^ splenocytes were counted and transferred into 1.5 mL tubes. The phycoerythrin-conjugated (PE) anti-CD4 (Thermo Scientific, Waltham, MA, USA; 0.125 µg), allophycocyanin-conjugated (APC) anti-CD8 (Thermo Scientific, Waltham, MA, USA; 0.125 µg), fluorescein-isothiocyanate-conjugated (FITC) anti-CD3 (Thermo Scientific, Waltham, MA, USA; 0.25 µg), and PE anti-F4/80 monoclonal antibodies (Thermo Scientific, Waltham, MA, USA; 0.5 µg) were added to the splenocytes. The mixtures were incubated for 2 h on ice in the dark. The antibody-stained splenocytes were washed with 0.5 mL of PBS prior to centrifugation at 2000× *g* for 5 min. Subsequently, the antibody-stained splenocytes were fixed with 1% (*w*/*v*) paraformaldehyde in PBS before being kept at 4 °C. A flow cytometer (BD FACSCanto II Flow Cytometer, Erembodegem, Belgium) was used to analyze the splenocytes and data were analyzed with the BD FACSDiva™ software version 6.1.2 (BD Biosciences, San Jose, CA, USA).

### 4.14. Immunogenicity of the Chimeric VLPs

The sera collected from the mice on the second, fifth, eight, and ninth weeks were analyzed using ELISA. The RBDs of the β, δ, and Wu variants of SARS-CoV-2 were expressed and purified as described above. The purified β-, δ- and Wu-RBDs of SARS-CoV-2 were immobilized on a 96-well microtiter plate overnight at 4 °C. Excess proteins were removed by washing three times with TBST [0.01% (*v/v*) tween-20, 50 mM Tris-HCl, 150 mM NaCl; pH 7.4] before blocking with 1x milk-diluent (KPL, Milford, MA, USA; 200 µL) for 1 h at 25 °C. The wells were then washed again as above before adding serum samples (1:5000 dilution in TBS; 100 µL) and incubated for 2 h at 25 °C. Next, the wells were washed again as above before incubating with the alkaline-phosphatase-conjugated anti-mouse antibody (1:5000 dilution in TBS; 100 µL) for 2 h at 25 °C. Lastly, the wells were washed again as above before adding p-nitrophenyl phosphate (100 µL per well), and incubated for 20 min in the dark for color development. The absorbance at a wavelength of 405 nm was measured with a microtiter plate reader (BioTek, Winooski, VT, USA).

### 4.15. Quantification of Cytokines

The concentrations of cytokines (IFN-γ, TNF-α, IL-5, IL-6, and IL-12p70) in mouse peripheral blood were determined using a Mouse Magnetic Luminex^®^ Screening Assay Kit (8-Plex) (R&D Systems, Minneapolis, MN, USA) based on the standard protocol. Plates were read with Luminex^®^ 200 (R&D Systems, Minneapolis, MN, USA).

### 4.16. Statistical Analysis

The variations of antibody titers, cytokine quantification, and immunophenotyping were statistically analyzed using a one-way ANOVA. Duncan’s multiple-range method was applied to differentiate significant differences among different groups. *p*-values of less than 0.05 and 0.001 are considered significant and very significant, respectively. All data analysis was performed with SPSS statistics software (IBM Corporation, Armonk, NY, USA).

## 5. Conclusions

In conclusion, the recombinant plasmids harboring the coding sequences of CΔ116-*Mr*NV-CP^β-RBD^ and CΔ116-*Mr*NV-CP^δ-RBD^ were successfully constructed and introduced into BL21 (DE3) competent cells. DLS and STEM showed that the *E. coli*-expressed CΔ116-*Mr*NV-CP^β-RBD^ and CΔ116-*Mr*NV-CP^δ-RBD^ self-assembled into a homogeneous population of spherical VLPs. The immunogenicity of the VLPs of CΔ116-*Mr*NV-CP^β-RBD^ and CΔ116-*Mr*NV-CP^δ-RBD^ was studied in BALB/s mice, and the result demonstrated the potential of these VLPs as SARS-CoV-2 vaccine candidates due to their ability to elicit humoral and cellular immune responses. Although the immunological study suggests that the VLPs of CΔ116-*Mr*NV-CP^β-RBD^ and CΔ116-*Mr*NV-CP^δ-RBD^ are potential COVID-19 vaccines, a virus challenge study in animal models will provide insights into the protective efficacy of the chimeric VLPs against SARS-CoV-2 infection.

## 6. Patents

A patent entitled “COVID-19 vaccine based on *Macrobrachium rosenbergii* nodavirus platform” (Patent application no: PI 2021007654) was filed on 22 December 2021.

## Figures and Tables

**Figure 1 ijms-24-04398-f001:**
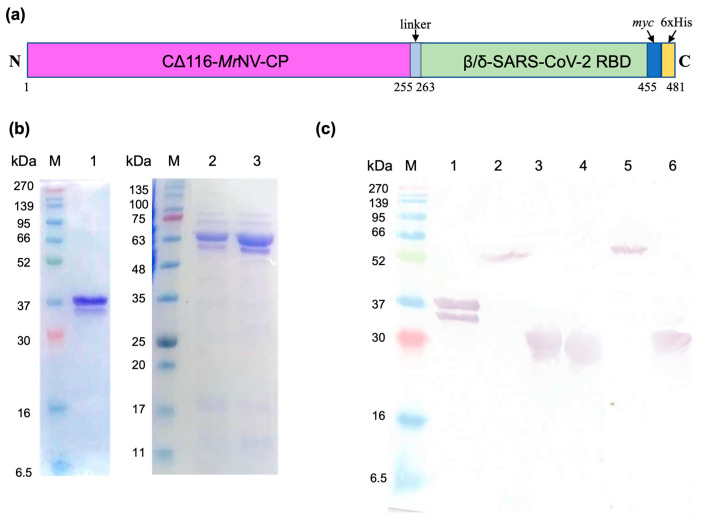
SDS-PAGE and Western blot analysis of the recombinant proteins. (**a**) A schematic representation of β- or δ-SARS-CoV-2 RBDs fused to the C-terminus of the protruding (P) domain truncated-*Mr*NV-CP (CΔ116-*Mr*NV-CP). The *myc* epitope and a 6xHis-tag at the C-terminal region are labeled. The positions of the amino acid residues of each domain are numbered. (**b**) SDS-PAGE analysis of purified (lane 1) CΔ116-*Mr*NV-CP with a molecular mass of ~33 kDa, (lane 2) β-SARS-CoV-2 RBD fused to the C-terminus of CΔ116-*Mr*NV-CP (CΔ116-*Mr*NV-CP^β-RBD^), and (lane 3) δ-SARS-CoV-2 RBD fused to the C-terminus of CΔ116-*Mr*NV-CP (CΔ116-*Mr*NV-CP^δ-RBD^). Lanes 2 and 3 show that the molecular mass of the purified protein is ~55 kDa. Purified proteins were then analyzed with Western blotting. Lanes 1 to 6 in (**c**) show that CΔ116-*Mr*NV-CP, CΔ116-*Mr*NV-CP^β-RBD^, β-RBD alone, Wu-RBD alone, CΔ116-*Mr*NV-CP^δ-RBD^, and δ-RBD alone, respectively, were detected by the anti-His monoclonal antibody. Protein markers in kilo Dalton (kDa) are indicated on the gels and blot.

**Figure 2 ijms-24-04398-f002:**
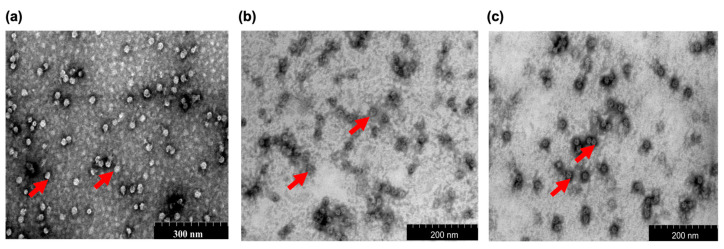
Scanning transmission electron microscopic (STEM) analysis of the purified recombinant proteins. Micrographs show that (**a**) protruding domain truncated-nodavirus capsid protein (CΔ116-*Mr*NV-CP), (**b**) CΔ116-*Mr*NV-CP fused with β-SARS-CoV-2 RBD (CΔ116-*Mr*NV-CP^β-RBD^), and (**c**) CΔ116-*Mr*NV-CP fused with β-SARS-CoV-2 RBD (CΔ116-*Mr*NV-CP^δ-RBD^) assembled into spherical VLPs. Protein samples were negatively stained with 2% (*w*/*v*) of uranyl acetate, and viewed with 200,000× magnification as shown by the scale bar. The red arrows indicate VLPs formed.

**Figure 3 ijms-24-04398-f003:**
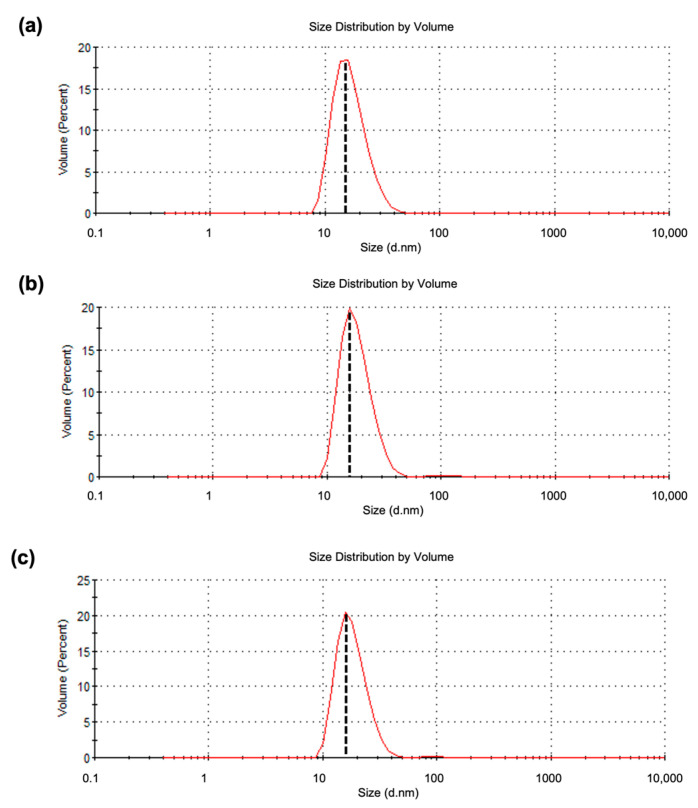
Dynamic light scattering analysis of the purified recombinant proteins. (**a**) CΔ116-*Mr*NV-CP, (**b**) CΔ116-*Mr*NV-CP harboring the β-SARS-CoV-2 RBD (CΔ116-*Mr*NV-CP^β-RBD^), and (**c**) CΔ116-*Mr*NV-CP harboring the δ-SARS-CoV-2 RBD (CΔ116-*Mr*NV-CP^δ-RBD^). The vertical dotted lines intersecting the peaks represent the average diameter of VLPs derived from the CΔ116-*Mr*NV-CP, CΔ116-*Mr*NV-CP^β-RBD^, and CΔ116-*Mr*NV-CP^δ-RBD^ with hydrodynamic diameters of 16.99, 18.40, and 18.39 nm, respectively.

**Figure 4 ijms-24-04398-f004:**
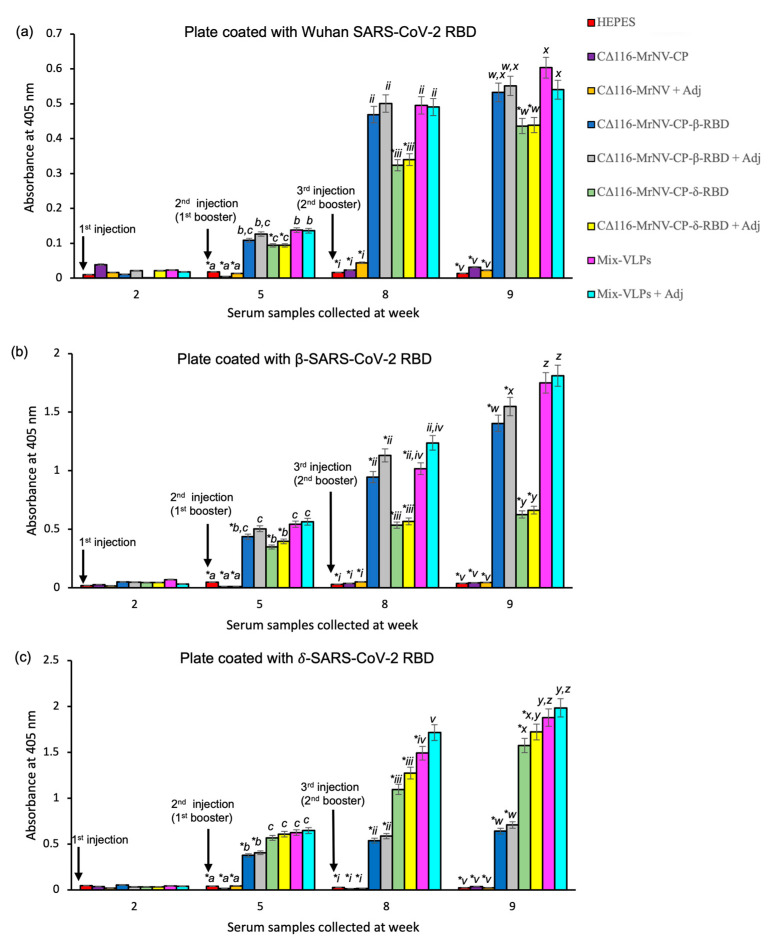
Immunogenicity analysis of different formulations of chimeric VLPs in BALB/c mice. Total antibodies against the (**a**) ancestral Wuhan, Wu-, (**b**) β-, and (**c**) δ-SARS-CoV-2 RBD in the sera collected at weeks 2 (before injection), 5 (3 weeks after 1st injection), and 8 (3 weeks after 1st booster), and 9 (1 week after 2nd booster) were analyzed. The immunized sera were used to determine the presence of antibodies against the β, δ and Wu-SARS-CoV-2 RBDs coated on the microtiter plate wells. Statistical significance (*p* < 0.001) is denoted by letters and Roman numerals shown above the bars. Insignificant differences are indicated by the same letter. Asterisks (*) denote significant differences (*p* < 0.001) compared to the animal group receiving the Mix-VLPs (a mixture of CΔ116-*Mr*NV-CP^β-RBD^ and CΔ116-*Mr*NV-CP^δ-RBD^ VLPs) with or without adjuvant. The standard deviations of triplicate measurement are represented by error bars.

**Figure 5 ijms-24-04398-f005:**
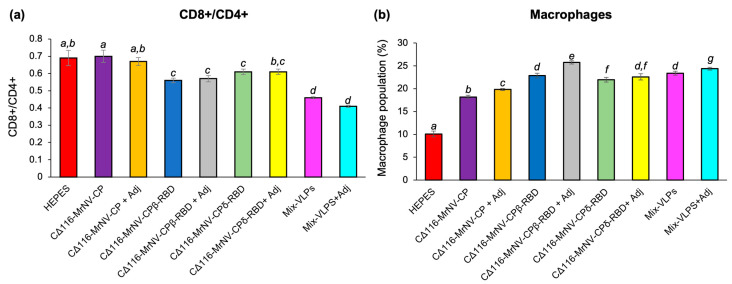
Immunophenotyping of mouse splenocytes. (**a**) The ratio of CD8^+^/CD4^+^, and (**b**) the percentage of macrophage isolated from the mouse splenocytes. Statistical significance (*p* < 0.001) is denoted by italic letters shown above the bars. Insignificant differences are indicated by the same letter. Standard deviations of triplicate measurements are indicated by error bars.

**Figure 6 ijms-24-04398-f006:**
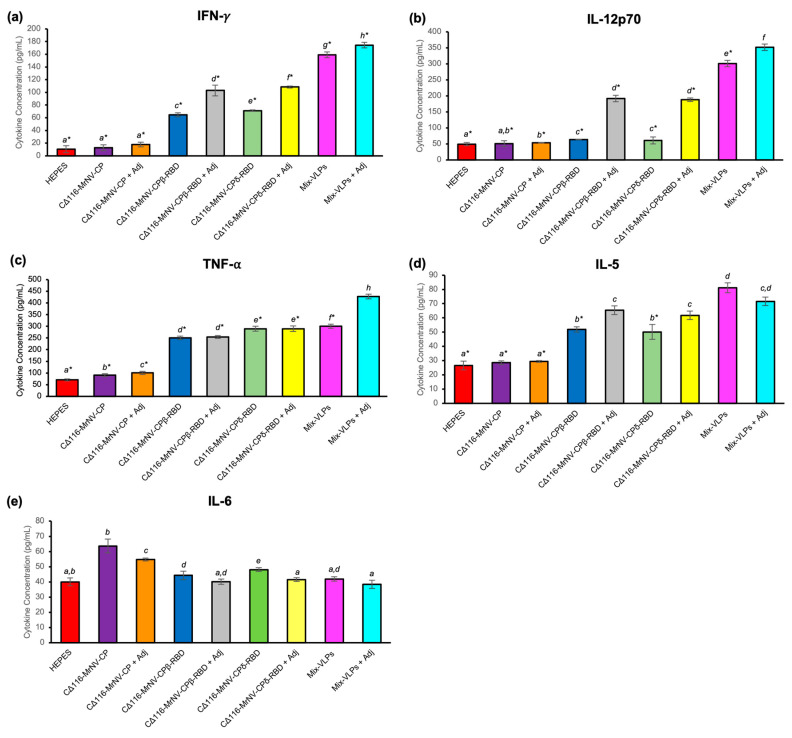
Multiplex quantification of cytokines in the sera of mice immunized with the chimeric VLPs. Different types of cytokines, (**a**) IFN-γ; (**b**) IL-12p70; (**c**) TNF-α; (**d**) IL-5; and (**e**) IL-6, collected from sera of the study groups, were quantitated. The negative groups were inoculated with HEPES buffer or the VLPs of CΔ116-*Mr*NV-CP with and without the AddaVax adjuvant (Adj), while the test groups were inoculated with the VLPs of CΔ116-*Mr*NV-CP displaying the β-SARS-CoV-2 RBD (CΔ116-*Mr*NV-CP^β-RBD^) and CΔ116-*Mr*NV-CP displaying the δ-SARS-CoV-2 RBD (CΔ116-*Mr*NV-CP^δ-RBD^) and a mixture of CΔ116-*Mr*NV-CP^β-RBD^ and CΔ116-*Mr*NV-CP^δ-RBD^ (Mix-VLPs) with and without the AddaVax adjuvant. Statistical significance (*p* < 0.001) is denoted by italic letters shown above the bars. Insignificant differences are indicated by the same letter. The asterisks indicate significant difference (*p* < 0.001) by comparing the concentration of cytokines produced in the test groups with that from the mice immunized with Mix-VLPs supplemented with adjuvant. Standard deviations of triplicate measurements are shown by error bars.

**Table 1 ijms-24-04398-t001:** Immunophenotyping of mouse splenocytes.

Groups	Percentage of Cell Gated (%)	
	CD3^+^CD4^+^	CD3^+^CD8^+^
HEPES	9.27 ± 0.46 ^a^	6.40 ± 0.1 ^i^
CΔ116-*Mr*NV-CP	10.50 ± 0.20 ^b^	7.36 ± 0.21 ^ii^
CΔ116-*Mr*NV-CP+ Adj	14.37 ± 0.15 ^c^	9.57 ± 0.35 ^iii^
CΔ116-*Mr*NV-CP^β-RBD^	18.37 ± 0.15 ^d^	10.30 ± 0.26 ^iii,iv^
CΔ116-*Mr*NV-CP^β-RBD^+ Adj	18.40 ± 0.17 ^d^	10.63 ± 0.25 ^iv^
CΔ116-*Mr*NV-CP^δ-RBD^	17.40 ± 0.36 ^d,e^	10.53 ± 0.15 ^iv^
CΔ116-*Mr*NV-CP^δ-RBD^+ Adj	18.20 ± 0.35 ^e^	11.13 ± 0.32 ^iv^
Mix-VLPs	22.40 ± 0.17 ^e^	10.40 ± 0.10 ^iv^
Mix-VLPs + Adj	26.30 ± 0.30 ^f^	11.00 ± 0.30 ^iv^

All data were of statistical significance (*p* < 0.001). Immunophenotyping of the splenocytes of mice injected with buffer; VLPs of CΔ116-*Mr*NV-CP, CΔ116-*Mr*NV-CP displaying the β-SARS-CoV-2 RBD (CΔ116-*Mr*NV-CP^β-RBD^), and δ-SARS-CoV-2 RBD (CΔ116-*Mr*NV-CP^δ-RBD^); and a mixture of VLPs (CΔ116-*Mr*NV-CP^β-RBD^ and CΔ116-*Mr*NV-CP^δ-RBD^; Mix-VLPs), with and without the AddaVax adjuvant (Adj), was performed and tabulated. The fractions of Th cells and CTL populations are indicated by the percentages of CD3^+^CD4^+^ and CD3^+^CD8^+^, respectively. Superscripted letters and Roman numerals represent significantly different values (*p* < 0.001), while those with the same letters are insignificantly different.

**Table 2 ijms-24-04398-t002:** PCR primers.

Primer	Nucleotide Sequence 5′-3′
Forward Wuhan	5’GACAGCCATGGCCAATA TTACAAACTTGTGCCC-3’; NcoI restriction site is underlined
Reverse Wuhan	5’CTGATAAGCTTCTCCACAAACAGTTGCTGGTG-3’; HindIII restriction site is underlined
Forward Beta/Delta	5’TAAACCATGGCCCTTAATATTACAAACTTGTGCCCTTTT-3’; NcoI restriction site is underlined
Reverse Beta/Delta	5′AGCTTCGAATTCAACAGTTGCTGGTGCATGTAGAAG-3’; EcoRI restriction site is underlined

## Data Availability

Not applicable.

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
