# Peer review of "Virus-like Particles of Nodavirus Displaying the Receptor Binding Domain of SARS-CoV-2 Spike Protein: A Potential VLP-Based COVID-19 Vaccine"

_ijms, 2023, doi:10.3390/ijms24054398_

Round 1
Reviewer 1 Report
Good morning
for the all Authors,
Analyzing the Manuscript with ID: ijms-2107693- entitled "Virus-like Particles (VLP) of Nodavirus Displaying the Receptor Binding Domain of SARS-CoV-2 Spike Protein: A potential VLP-based COVID-19 Vaccine" for a possible publication in International Journal of Molecular Sciences – MDPI (ISSN: 1422-0067; IF=6.208).
As a result:
1. The article follows all the specific instructions of the journal presented in: aims and scope, instructions for authors and other information about the journal.
2. The data presented in this manuscript are well structured and coherent;
3. The methods, statistical analysis and results are well presented and easy to understand;
4. The bibliography chosen by the authors corresponds to the requirements and refers to the subject of the article.
In conclusion:
I accept the article.
I recommend publishing the article in your prestigious journal.

Author Response
We thank reviewer 1 for reviewing our manuscript and recommending our manuscript to be accepted for publication.
Reviewer 2 Report
Part 4 (Materials and methods) should be part 2 and than follow results, discussion and conclusion
Author Response
Thank you for the suggestion. The format of the manuscript is based on the MDPI LaTeX template downloaded from the submission menu.
Reviewer 3 Report
Virus-like particles (VLPs) are an attractive platform for developing vaccines against infectious diseases such as COVID-19, due to their impressive versatility and immunological applications. In the manuscript entitled “Virus-like Particles (VLP) of Nodavirus Displaying the Receptor Binding Domain of SARS-CoV-2 Spike Protein: A potential VLP-based COVID-19 Vaccine” the authors prepared a construct by fusing the RBDs of the Beta (β) and Delta (δ) variants of SARS-CoV-2 with the protruding domain truncated Macrobrachium rosenbergii noda-virus capsid protein (CP) after which the proteins were expressed in E. coli. Next, the immunogenicity, immunophenotyping, and cytokine profiles of these recombinant proteins (CΔ116-MrNV-CPβ-RBD and CΔ116-MrNV-CPδ-RBD) were studied in BALB/c mice. This is an interesting and well-written manuscript. The data are clearly presented, the material and methods are fully described and the conclusions are compelling. There are, however, some major concerns related to vaccine neutralization that need to be addressed before the manuscript will be accepted for the publication;
Comments:
1) In this study authors determine the antibodies against the RBDs using ELISA. The antibody level measurements do not give any information about the protective effect of any vaccine. For most viruses, the best correlate of protection upon vaccination is viral neutralization. In this study, authors need to show the capacity of the induced antibodies in neutralizing wild-type SARS-CoV-2 along with β- and δ-variants along with the antibody avidity measurements. These experiments are important and will further support and strengthen their findings.
2) It would be great to perform a competitive assay to determine whether the anti-RBD sera from immunized mice can compete with ACE2 for binding to the RBD.
3) It would be interesting to know the effect of these recombinant proteins against the new continuously evolving VOCs of SARS-CoV-2, such as omicron variants.
Author Response
Comment 1: In this study authors determine the antibodies against the RBDs using ELISA. The antibody level measurements do not give any information about the protective effect of any vaccine. For most viruses, the best correlate of protection upon vaccination is viral neutralization. In this study, authors need to show the capacity of the induced antibodies in neutralizing wild-type SARS-CoV-2 along with β- and δ-variants along with the antibody avidity measurements. These experiments are important and will further support and strengthen their findings.
Response: Thank you for the suggestion. We have thought of performing pseudovirus/plaque reduction neutralizing assays (P/PRNA) with pseudo- or live virus to demonstrate the capability of the induced antibodies in neutralizing SARS-CoV-2. We have discussed this additional experiment with our university’s biosafety committee, but was rejected because the university does not have biosafety level III laboratory for handling pseudo- or live virus. We have approached other institutions which have biosafety level III laboratories, but their concern is to obtain approval from their university’s biosafety committee and the national biosafety committee, which may take at least 6 months to one year for approval. The main aim of current manuscript is to prove the concept that the truncated MrNV-CP can be used to display the RBD of β- and δ-SARS-CoV-2, as well as to study their immunogenicity in BALB/c mice.
Comment 2: It would be great to perform a competitive assay to determine whether the anti-RBD sera from immunized mice can compete with ACE2 for binding to the RBD.
Response: In this study, the chimeric VLPs displaying the RBDs were inoculated into the mice via subcutaneous injection. The competitive assay with ACE2 is more applicable if animals were immunized via intranasal or intratracheal (or both) routes. The studies that support this finding include: i) Dalvie et al., Sci. Adv.8, eabl6015 (2022), ii) Sano et al., Nature Comm, 13(1):5135 (2022).
References:
- Dalvie, N. C., Tostanoski, L. H., Rodriguez-Aponte, S. A., et al. (2022). SARS-CoV-2 receptor binding domain displayed on HBsAg virus-like particles elicits protective immunity in macaques. Sci Adv. 8(11):eabl6015. doi: 10.1126/sciadv.abl6015. Epub 2022 Mar 16.
- Sano, K., Bhavsar, D., Singh, G., et al. (2022). SARS-CoV-2 vaccination induces mucosal antibody responses in previously infected individuals. Nature communications, 13(1), 5135.
Comment 3: It would be interesting to know the effect of these recombinant proteins against the new continuously evolving VOCs of SARS-CoV-2, such as omicron variants.
Response: Thank you for the suggestion. It is interesting to know how the effect of these recombinant proteins against new variants such as Omicron although the RBD derived from previous VOCs has been shown to have lower neutralizing sensitivity due of mutations in the S protein (Ai et al., Emerging microbes & infections, 11, 337–343 (2022). We did not include the RBD derived from Omicron VOCs because this variant was not discovered at the time the experiment was designed in mid-2021. Additional experiments would be unfeasible as a new batch of mice is required to analyze the immune response. Additionally, approvals by IACUC (institutional animal care and use committee), as well as the university and national biosafety committees must be obtained prior to the additional experiment. The application and approval process may take at least 6 months to one year.
In fact, the main aim of this study was to prove the concept that truncated MrNV-CP can be used to display the RBD of β- and δ-SARS-CoV-2, as well as to study their immunogenicity in BALB/c mice. We will include Omicron variants in future studies after obtaining approvals from the IACUC and biosafety committees.
Reference:
- Ai, J., Zhang, H, Zhang, Y., Lin, K., et al. (2022). Omicron variant showed lower neutralizing sensitivity than other SARS-CoV-2 variants to immune sera elicited by vaccines after boost. Emerging microbes & infections,11, 337–343.
Round 2
Reviewer 3 Report
The authors provided the explanation for most of the raised comments except for some of the suggested experiments which can further improve this manuscript. I agree that, in order to do some of the additional sets of experiments, authors need to have a biosafety level III laboratory. However, based on the author’s response, it seems that their university does not have a biosafety level III laboratory for handling pseudo- or live viruses, and obtaining collaborative approval from other institutions may take at least 6 months to one year.
Overall, this is an interesting and well-written manuscript with compelling conclusions. Considering all the facts and the author’s responses, I would like to recommend this article for publication.